# Engineering Interfacial Environment of Epigallocatechin Gallate Coated Titanium for Next-Generation Bioactive Dental Implant Components

**DOI:** 10.3390/ijms24032661

**Published:** 2023-01-31

**Authors:** Giorgio Iviglia, Marco Morra

**Affiliations:** Nobil Bio Ricerche srl, Via Valcastellana 26, 14037 Portacomaro, AT, Italy

**Keywords:** titanium abutment, surface functionalization, polyphenols, epigallocatechin, wettability, antioxidant power, surface characterization

## Abstract

In view of endowing the surface of abutments, a component of titanium dental implant systems, with antioxidant and antimicrobial properties, a surface layer coated with epigallocatechin gallate (EGCg), a polyphenol belonging to the class of flavonoids, was built on titanium samples. To modulate interfacial properties, EGCg was linked either directly to the surface, or after populating the surface with terminally linked polyethyleneglycol (PEG) chains, Mw ~1600 Da. The underlying assumption is that fouling-resistant, highly hydrated PEG chains could reduce non-specific bioadhesion and magnify intrinsic EGCg properties. Treated surfaces were investigated by a panel of surface/interfacial sensitive techniques, to provide chemico–physical characterization of the surface layer and its interfacial environment. Results show: (i) successful EGCg coupling for both approaches; (ii) that both approaches endow the Ti surface with the same antioxidant properties; (iii) that PEG-EGCg coated surfaces are more hydrophilic and show a significantly higher (>50%) interaction force with water. Obtained results build up a rationale basis for evaluation of the merits of finely tuning interfacial properties of polyphenols coated surfaces in biological tests.

## 1. Introduction

A major goal of biomedical implant devices science is to stimulate proper host tissue response through materials’ surface properties [1]. The fine tuning of surface chemistry allows control over interfacial chemical interactions and determines ensuing surface properties such as wettability and surface charge. Of particular interest, when it comes to biomedical devices, is the modification of surface chemistry through surface-immobilization of signalling biological molecules such as extracellular matrix proteins, relevant peptides, glycosaminoglycans, and growth factors [2,3,4,5]. Immobilization of functional molecules from the plant kingdom has also been the subject of several intriguing studies [6,7,8,9]. The underlying tenet is the endowment of the implant surface with the properties of the selected molecule, as related to its role in mechanisms presiding over tissue regeneration and repair.

A further refinement of plain surface immobilization of biologics as a way to provide biochemical stimulation of peri-implant tissue healing is the concurrent building up of a controlled interfacial chemical environment. This amounts to the structuring of actual composite material nanolayers on the surface, where the active biologic plays through specific interactions and further components concur to set the stage for favourable interfacial properties. By way of an example, it is well known that non-specific adsorption (fouling) is dependent on wettability of the surface, and/or on surface charge and that it can spoil the properties of bio-specific surfaces [10,11]. A possible strategy to circumvent this problem is the building up of a surface composite material nanolayer, whose chemistry involves both an active biological molecule and further chemical moieties intended for the control of wetting and charge.

We are involved with surface treatment of titanium dental implant systems and with the development of novel technologies to control tissue response at the titanium-host tissue interface. This paper deals with the chemical–physical description of the building up of a bioactive material nanolayer on top of a specific component of dental implant systems, which is the abutment. As described by Wennerberg et al. [12], dental implant systems in general consist of two components: implant and abutment. The abutment is fixed to the implant via a prosthetic screw and supports the dental restoration. It should have a surface that allows for tight soft-tissue adherence in the transmucosal zone to support peri-implant tissues and create a tight soft-tissue seal against bacteria thriving in the oral cavity. During the mucosa healing time, the surface should ideally be unfavourable to bacterial attachment and pathological biofilm formation, while keeping under control inflammatory response occurring on healing. Preferred present strategies to accomplish these goals involve the control of surface roughness of titanium abutments below 0.2 micrometres Ra or growth of the surface oxide layer through anodic oxidation. It has been reported that oxidized nanostructured titanium surfaces stimulate adhesion, proliferation, and extracellular matrix secretion of human gingival fibroblasts compared with machined surfaces [12,13,14,15].

Nanoengineering of the titanium surface through biologically active molecules, according to the principles described in the initial section, could further enhance abutment surface properties and long-term success of the dental implant system. A class of molecules from the plant kingdom appears ideally suited to fulfil the interfacial requirements of abutments: polyphenols, whose significance in biomedical applications is continuously rising, are well known antioxidant and possess antimicrobial properties [16,17,18,19,20,21,22]. They show very interesting behaviour at the solid–liquid interface and form continuous films, and they also present chemical moieties suitable for surface-immobilization [23,24,25]. An excellent recent review by Mele et al. describes existing methods for the building up of surface coatings of phenolic phytocompounds of medical interest [26]. Among them, here we focus on epigallocatechin gallate (EGCg), whose structure is shown in Figure 1, a widespread flavonoid contained, among others, in tea and grape skin. Beside sharing antioxidant and anti-inflammatory properties common to other flavonoids, EGCg antimicrobial activity has been widely documented [27,28,29]. Among others, it has been shown that EGCg is effective against *Streptococcus mutans*, a biofilm forming, Gram positive bacterium commonly found in oral cavities [30]. No general agreement exists on the reason behind EGCg (and other polyphenols) antimicrobial activity. The two most supported hypothesis involve either the generation of hydrogen peroxide on oxidation, or the intercalation in the lipid bilayer bacterial membrane, resulting in increased membrane permeability [30].

Endowing the titanium abutments surface with the properties of EGCg is of obvious interest. Surface modification of titanium through EGCg is straightforward, and it can be accomplished either by simple adsorption or by linking to spacers or chemical functionalities. Here we are not simply interested in EGCg coatings on titanium. Rather, we aim at building a specific functional material nanolayer, involving EGCg as an active player; we want to seat EGCg not on a hard, hydrophobic interface, prone to strong non-specific adsorption, but within a hydrated, hydrophilic, interfacial environment. Strong water–surface interaction should temperate the interfacial driving force for bacterial adsorption, magnifying the antimicrobial effect of EGCg. Polyethyleneglycol, whose -CH2-CH2-O- repeating unit has been widely discussed with respect to fouling resistance, is a suitable candidate to build up a strongly hydrated interfacial environment [31,32,33].

The experimental approach we adopt exploits Polydopamine (PDA) chemistry as an intermediate layer directly linked to titanium [34]. Amino-terminated polyethylene glycol (aPEG) is then linked to PDA through coupling reactions involving Schiff base formation or Michael addiction. Finally, EGCg is slipped inside the PDA-linked aPEG layer, exploiting residual coupling sites on PDA and resulting in a composite surface layer. The building up of the material nanolayer is followed by X-ray Photoelectron Spectroscopy (XPS), Atomic Force Microscopy (AFM), and zeta potential measurement. The activity of EGCg on the surface is measured by the Folin Ciocalteux method, and wetting measurement according to the Wilhelmy plate method is used to evaluate surface hydrophilicity. The aim of this work is to gather evidence of building up EGCg active surfaces endowed with different interfacial properties with respect to interaction with water as compared to simple EGCg immobilization, as a rational basis for planning of the evaluation of antimicrobial properties in future experiments.

## 2. Results

### 2.1. X-ray Photoelectron Spectra

XPS analysis allows one to follow the evolution of the chemistry of the surface nanolayer built on top of titanium along the different steps. The surface composition data of the tested samples are reported in Table 1. Figures obtained on Ti are typical of clean titanium surfaces. After PDA coating, the substrate signal almost disappear and the material surface chemistry turns organic. The detected stoichiometry is in excellent agreement with expected data, in particular the N/C ratio is 0.11, as reported in 30. This shows that a homogeneous PDA coating covers the pristine titanium surface.

Interpretation of subsequent steps must take into account the fact that molecules that are introduced are mostly based on carbon and oxygen and do not bear “markers”, that is, specific elements that could indicate successful coupling. However, both EGCg and aPEG either do not contain nitrogen (EGCg) or they contain it in limited amounts (aPEG). Therefore, the linking of these molecules to the surface layer should promote a decrease of the N/C ratio. Table 1 shows that this is indeed the case, both for direct EGCg coupling to PDA and for aPEG linking to PDA. A further decrease in the N/C ratio is observed when EGCg is coupled to Ti-PDA-aPEG, suggesting that, indeed, residual linking sites are available to capture EGCg on the PDA-aPEG surface layer.

Further support to successful coupling is offered by the evaluation of high-resolution C1s peaks of the samples. Results are shown in Figure 2. As reported in the Experimental section, all peaks have been referenced to the internal standard C-C component, whose binding energy is 285.0 eV, according to the usual XPS practice. Peak height has been normalized, so that for all peaks the maximum is 1. The C1s peak of PDA is broad because of multiple C-N, C-O functionalities giving rise to different, partially overlapping, components. The introduction on the surface layer of EGCg and/or aPEG should affect peak shape, in particular through the increase of the 286.5 eV component due to carbon single-bond oxygen, present in the repeating unit of aPEG and abundant in the EGCg molecular structure. Figure 2 shows, indeed, the progressive growth of the 286.5 eV component as C-O-containing moieties are introduced in the surface layer through aPEG or EGCg or both.

In summary, based on common chemical sense, XPS analysis data support the building up of a surface layer according to the expected course. A further piece of information that can be obtained by this technique is an estimate of the thickness of the surface composite layers, exploiting angle resolved XPS. Acquiring XPS spectra at different take off angles (that is the angle between the sample and the electron analyzer) affects the sampling depth of the technique, that is the thickness of the layer whose photoemitted electrons are analysed. While previous data were obtained using the standard take off angle of 45°, further spectra were obtained at grazing angle, that is 15°, and at 75°. In the former, case surface sensitivity is maximized, the probed depth is around 4 nm. Acquiring spectra at 75° take-off, instead, prompts acquisition of photoelectrons from a surface layer approximately 10 nm thick. The results obtained on Ti-PDA-aPEG-EGCg are reported in Figure 3. Two important observations are as follows: (i) At grazing angle, no signal from the substrate (Ti) is detected, while maximizing the sampling depth shows a small but significant Ti peak. (ii) The background of the spectrum, made up by secondary electrons, is very different at 15° and 75°, as nicely observed in the >600 eV spectral region. In the former case it has the typical linear shape found in organic materials, meaning that most collected photoelectrons come from an organic layer. Increasing sampling depth leads to a different background shape, because part of the signal comes from the underlying titanium layer. Shortly, angle resolved XPS suggests that the surface material layer is vertically non-homogeneous within the explored sampling depths, the substrate being detectable when sampling depth is increased. This indicates that the thickness of the surface composite organic layer is comparable with the thickest tested sampling depth, which is below 10 nm.

### 2.2. Antioxidant Power of EGCg Coated Samples

Previous surface chemistry data suggest successful coupling of EGCg to PDA-coated Ti for both selected approaches. To confirm these results, and to gather more information on the quantitative aspects of EGCg immobilization, the antioxidant power of the coated surfaces was measured through DPPH assay. Results are reported in Table 2. Data show the percent value of free radical reduced by a coated Ti cylinder. The first point to note is that both approaches yield Ti surfaces endowed with antioxidant power. This shows that EGCg is indeed linked to the Ti surface, confirming XPS indications. As to quantitative aspects, the data show that both coatings produce similar antioxidant power values: 13.3% for PDA-EGCg and 14.0% for PDA-PEG-EGCg. This result apparently suggests that coupling aPEG to PDA does not hinder further coupling of EGCg, either in terms of steric hindrance or in terms of reactive sites available on PDA. This topic will be discussed in some depth in the Discussion section.

HPLC analysis (Appendix A) on the released solution from 10 implants for each coating, showed no phenolic release, which means that the polyphenolic molecule EGCg exerts its potential from the surface of the implant and is not due to the releasing in the media.

### 2.3. Atomic Force Microscopy

Previous chemical data indicate that EGCg is successfully linked to the PDA-coated surface for both selected approaches, endowing the surface with antioxidant properties. Morphology changes due to the building up of the antioxidant surface material layer was followed by non-contact AFM. Morphology changes due to the building up of the antioxidant surface material layer was followed by non-contact AFM. Because our goal is to evaluate subtle differences between the two approaches, AFM measurements were performed on freshly cleaved atomically flat mica surfaces, to take full advantage of the exceptional vertical resolution of the technique. Machined Ti shows Sa values of 200–300 nm, preventing every speculation about morphological changes due to the building up of the composite surface layer. Therefore, data were obtained and are presented under the assumption that what happens on PDA-coated mica reflects what happens on PDA-coated Ti. This assumption sounds reasonable, because PDA shows similar coating efficiency towards metal, ceramics, and polymers [34]. The main results are reported in Table 3 and Figure 4. Analysis of the substrate (mica) yields the expected results, that is an atomically flat surface. This is quantitatively confirmed in Table 3 by the vertical roughness parameter (Sa) and, most of all, by the hybrid roughness parameter (Sdr), which shows that the actual surface is practically coincident with the geometrical one. Shortly, no topographical features exist in the analysed area at this level of resolution.

After PDA coating, Figure 4a, the typical spot morphology is observed on the surface, quite uniformly distributed. SEM images confirm the presence of spherical particles on the PDA surface. Beside the increase of the Sa value, a significant increase of the surface area is detected through the Sdr value, due to the nanometre-size topographical features projecting over the surface. Adding aPEG slightly modifies the morphology of the surface; apparently larger agglomerates build-up on the surface. The addition of EGCg on PDA-aPEG apparently promotes more significant changes as compared to EGCg addition to plain PDA, suggesting that, in the former case, reorganization of the surface architecture occurs beside EGCg coupling.

### 2.4. ζ-Potential Measurement

Further insights on the nature of the surface layers built on titanium were gathered by electrokinetic effects at the solid–liquid interface, evaluated through *ζ*-potential measurement. In particular, streaming current was measured as a function of pH, in the 2.5–9.5 range, in 1 mM KCl solution. The contact between a solid surface and a water-based medium leads to the development of a surface charge at the interface. This charge is one of the surface characteristics which could affect the interaction between the material and the biological environment. The pH of an aqueous solution is the driving force for the building up of surface charge, through acid-base dissociation of chemical functionalities eventually available on the solid surface; at high pH values, the dissociation of acidic groups will be enhanced, while the protonation of basic groups will be suppressed, and vice versa. Even surfaces that do not have acid-base groups show a pH dependent behaviour of surface charge, not dictated by the acid-base dissociation of chemical functionalities, but mostly by adsorption of ions contained in the solution (OH^−^, H_3_O^+^, Cl^−^, K^+^) as a function of pH. Results are shown in Figure 5. In the case of Uncoated Ti, the pH scan is typical of a very weak acid-base interfacial activity, and resulting potential is driven by the pH dependent adsorption of ions. The surface charge on PDA coating is probably due to quinone imine and catechol groups. It has been reported in the literature that the overall charge of PDA coatings is negative, although there is no uniformity regarding the zeta potential value because this depends on many factors (pH, electrolyte).

In particular, at high pH, the negative surface charge may originate from the dissociation of quinone imine and catechol groups. An inflection point is present around pH 6, which correspond to the average pKa of quinone imine. At lower pH, the surface charge becomes less negative, and this may be due to the protonation of nitrogen atoms from the indole group. Furthermore, it has been reported in the literature that PDA could interact with aqueous solutions through various mechanism due to its molecular structure; if PDA binds through OH^−^ groups from quinone, it leaves more amine groups exposed to the aqueous environment and thus may provoke a less negative surface charge.

The coupling of aPEG and EGCg, or both, yields similar effects on the pH scan curve, substantially masking the features due to PDA. The obtained curves are generally similar; they reflect the lack of significant acid-base activity, as expected. EGCg is a catechin which is characterized by multiple phenolic hydroxyls. Either directly coupled to PDA, or slipped inside the PDA-linked aPEG layer, exposure of -OH groups yields slightly more negative surface charge compared to the remaining samples surface.

### 2.5. Wettability Measurements

The results of contact angle measurement are reported in Figure 6. Ti-PDA shows the expected characteristics of a mildly hydrophobic surface. The advancing angle is in good agreement with the results of Wang et al. [35]. The receding angle value indicates that, while polar groups are present on the surface, they are not so hydrophilic to “pin” the receding water front, in agreement with PDA structure. Introducing EGCg on the surface does not significantly affect the advancing angle. Significantly, the receding angle drops close to zero. These results are in agreement with the “classic” view of the wettability of chemically heterogeneous surfaces, whose advancing angle reflects the hydrophobic moieties, and the receding one the hydrophilic moieties, on the surface [36,37]. Therefore, the drop of the receding angle is explained by the pinning of the receding water front by the hydrophilic phenol moieties of EGCg.

Significant lowering of both advancing and receding angle, with respect to Ti-PDA, is obtained by coupling hydrophilic aPEG. The PEG repeating unit interacts strongly with water molecules, and this feature is at the basis of the widespread use of PEG in fouling-resistant coatings. This result confirms XPS indication of successful coupling of the amino-terminated PEG to PDA. The further coupling of EGCg to Ti-PDA-aPEG partially reverses the hydrophilic gain provided by PEG, as shown by the increase of the advancing angle. This result reflects the introduction on the surface of the relatively more hydrophobic EGCg with respect to PEG. The measured advancing angle, however, is well below the value measured on the surface, where EGCg is simply linked to PDA. The picture is consistent with that of a chemically heterogeneous surface, whose wetting properties are dictated by a mixture of PEG and EGCg molecular features.

Figure 7 compares the wetting loops obtained by Wilhelmy plate measurements on the two approaches to EGCg immobilization on Ti. The interaction between water and the surface, as detected by the force at zero depth of immersion (ZOI), increases in excess of 50% on Ti-PDA-aPEG-EGCg with respect to plain Ti-PDA-EGCg. This result confirms that, while both surfaces bear the same amount of active antioxidant EGCg, these molecules reside in different chemical environments, and this affects interfacial interactions and properties.

## 3. Discussion

Surface properties of dental implant devices are most often discussed in terms of osteointegration, which is the role of the surface on cells and biological mechanisms leading to intimate apposition of newly formed bone to the implant surface [38]. For osteointegration to occur, however, proper soft tissue healing and build-up of an impervious soft tissue seal that isolates the submerged bony implant environment from the oral cavity is required. Actually, the main weak-link for the long-term success of dental implants is not the stability of the bone–implant interface, but bacterial accumulation at the soft tissue interface and peri-implant tissue destruction because of side effects ensuing from host defence system response to bacteria. Counteracting bacterial adhesion on healing and promoting healthy mucosal healing are thus the main clinical requests of abutments, the components of dental implant systems directly facing the oral cavity [39].

Phenolic phytocompounds are emerging more and more as potential active players in many tissue regeneration mechanisms, and they also present intriguing anti-microbial properties [26,27,28,30]. Thus, they are logical candidates in the development of next generation implant systems abutments, endowed with finely tuned surface properties stimulated by bioactive molecules.

We want to test the biological effect of phenolic-based coatings in mechanisms of relevance to titanium abutments function. We hypothesize that the tuning of the interfacial environment on which they are seated can affect their response. Thus, we are interested not only in surface immobilization of phenolics, but also in the understanding of whether there is merit in the chemical nanoengineering of the abutment surface to provide a linking substrate optimized for the exploitation of their properties. To this end, in-depth chemico–physical characterization is required as a prerequisite to targeted biological testing.

In this work, EGCg, a well-known antioxidant flavonoid endowed with antimicrobial properties [27,28,30], is seated in two different interfacial environments, both exploiting the PDA chemistry to link it to the titanium surface. The first one involves plain EGCg linked to PDA. The second approach places EGCg amidst previously-linked aPEG chains. In particular, the PEG chains we used contain approximately 36 repeating units. These chains are terminally linked at the amino end to the PDA substrate. Therefore, one can envisage two definitely different interfacial structures confronting the host aqueous milieu: plain EGCg coupled to PDA leads to an impervious interface, with antioxidant molecules seated on PDA, building up a composite, moderately hydrophilic interfacial environment. When aPEG is pre-coupled, long hydrophilic chains, made up of approximately 36 repeating units whose molecular features strongly interact with water molecules, project from the surface into the interfacial environment. The presented surface chemistry data suggest that there is space enough between these terminally anchored chains for EGCg molecules to diffuse inside and reach reactive coupling sites on PDA. The resulting composite surface material layer shows the same antioxidant properties obtained on plain PDA coupling, but it interacts more strongly with water. Wilhelmy plate experiments show an increase of approximately 50% of the capillary force exerted at water–surface contact (Figure 7).

A first point of discussion is regarding the evidence of a similar load of antioxidant EGCg molecules on plain Ti-PDA and Ti-PDA-aPEG. One could expect that, because of the previous coupling of aPEG, either steric hindrance or exploitation of PDA reactive sites, or both, would decrease the yield of the EGCg coupling reaction to the surface. This is apparently not the case. In this respect, Wang et al. coupled amino-terminated PEG (Mw 5000 Da) to PDA-coated 316L stainless steel [35]. The experiments involved coupling both to plainly adsorbed PDA and to PDA after 1 h oxidation at 150 °C. The Authors report that coupling to oxidized PDA resulted in doubling of the amount of surface-linked aPEG with respect to coupling to non-oxidized PDA, aPEG density increasing from 368.5 to 752.7 ng/cm^2^. The underlying reason, according to Wang at el., is the modification of the surface chemistry of PDA on oxidation and the ensuing shift from a cathecol-rich to a quinone-rich substrate. In the present context, this suggests that the coupling of amino-terminated PEG to pristine PDA (i.e., without further oxidation) could target the fraction of existing quinones and it does not involve extensive surface coverage. Based on Wang et al.’s data, this amino-pristine PDA coupling leaves at least one half of the surface available, to the point that the density of linked molecules can be doubled when more quinones are made available by oxidation. Therefore, it is reasonable to conclude that the Ti-PDA-aPEG surface is not so crowded by PEG chains to exclude EGCg molecules from contact with the reactive sites on the PDA surface, hence the negligible effect of previously linked aPEG on ECG surface density, hence antioxidant power. A further point is that the present results suggest that aPEG and EGCg exploit different coupling sites on PDA, so that the aPEG-decorated PDA is still completely available for EGCg coupling. Indeed, Foss et al. showed that the pristine PDA coating contains a high concentration of free primary amines, and these sites are obvious target for EGCg, but not for aPEG [40]. Again, comparison between the antioxidant activity of Ti-PDA-EGCg and Ti-PDA-aPEG-EGCg suggests that the limiting factor for EGCg coupling is the availability of linking sites, not steric hindrance (or else more EGCg molecules would be coupled to Ti-PDA with respect to Ti-PDA-aPEG).

In this respect, it is tempting to interpret AFM data as the effect of surface reorganization after the different coupling steps. Actually, Figure 4 and Table 3 show a marked increase of both vertical (Sa) and hybrid (Sdr) roughness parameters on PDA-aPEG-EGCg, with respect to the other samples. Based on previous discussion, this could reflect the physical event that both EGCg and PEG, singly coupled to PDA, occupy only a fraction of the surface, as dictated by the density of the relevant coupling sites. Pictorially, there is ample room for these molecules to lie on the surface. Instead, when EGCg is coupled to the PDA-aPEG surface, the surface density of the coupled molecules obviously increases, and slipping of EGCg molecules among the terminally linked aPEG chains and ensuing surface crowding could force the latter to adopt a more “vertical”, more constrained conformation, a nanoarchitectural change reflected in the increase of the roughness parameters.

Previous results confirm the unique properties of PDA as a platform for molecular surface coupling and interfacial nanoengineering. The availability of coupling sites with different reactivity allows, as in the present case, the building up of finely tuned material nanolayers containing molecules targeted for different and specific purposes. This is obtained through facile, sequential steps. The remaining question is whether reported chemico–phyisical modifications of the interfacial nanoarchitecture can have effects on relevant biological properties, in particular bacterial attachment and pathological biofilm formation. The underlying assumption is that the 36-CH_2_-CH_2_-O- repeating units of terminally linked aPEG, protruding as highly hydrated chains in the interfacial milieu, could weaken the driving force for bacterial adsorption [32,33], supporting and amplifying EGCg antimicrobial activity. The soundness of this assumption will be evaluated in forthcoming dedicated experiments.

## 4. Materials and Methods

### 4.1. Sample Preparation

Polydopamine (PDA), amino-terminated polyethylene glycol (aPEG), epigallocatechin gallate (EGCg), and all other chemicals were purchased from Sigma Aldrich (Merck KGaA, St Luis, MI, USA).

Experiments were performed by building up the surface material layer on Ti samples of different size and shape, fit for the intended analysis. In particular, commercially pure (Cp) grade 4 Ti disks, 6 mm in diameter, were used for XPS analysis. Titanium plates, 1 × 2 cm, 0.2 mm thickness, were used for ζ-potential measurement. The same plates, cut as squares, 2.4 cm a side, were used for preparation of the samples intended for wetting measurements. The measurement of anti-oxidant power was performed on Ti cylinders, 13 mm height and 4 mm in diameter. For AFM analysis, samples were prepared both on Ti disks and on freshly cleaved mica, as reported in the relevant section of Results.

Experimental procedure for samples preparation was as follows:

PDA coating of freshly plasma-cleaned Ti samples was performed by soaking in PDA solution at a concentration of 1 mg/mL in Trizma Buffer pH 8.5 for 12 h, then rinsing with ultrapure water 5 times (coded Ti-PDA). A set of samples (coded Ti-PDA-aPEG) was then soaked in a solution of aPEG dissolved at the concentration of 0.5 mg/mL in Trizma Buffer pH 8.5 for 4 h, then rinsed with ultrapure water 5 times. EGCg coupling to either Ti-PDA (coded Ti-PDA-EGCg) or Ti-PDA-aPEG (coded Ti-PDA-aPEG-EGCg) was performed by soaking samples for 12 h in a solution 0.01 M of EGCg dissolved in ultrapure water. Finally, samples were rinsed 5 times with ultrapure water. All samples were dried in a controlled oven at 37 °C for 12 h before each analysis.

### 4.2. X-ray Photoelectron Spectra

X-ray photoelectron spectroscopy (XPS) analysis was performed using a Perkin Elmer PHI 5600 ESCA system (PerkinElmer Inc., Waltham, MA, USA). The instrument was equipped with a monochromatized Al anode operating at 10 kV and 200 W. The diameter of the analysed spot was approximately 500 μm; the analysed depth was approximately 5 nm. The base pressure was maintained at 10–8 Pa. The angle between the electron analyser and the sample surface was 45°. Analysis was performed by acquiring wide-range survey spectra (0–1000 eV binding energy) and detailed high-resolution peaks of relevant elements. Quantification of elements was accomplished using the software and sensitivity factors supplied by the manufacturer. High-resolution C1s peaks were acquired using a pass energy of 11.75 eV and a resolution of 0.100 eV/step.

### 4.3. Atomic Force Microscope

Atomic force microscopy (AFM) was used to explore the surface nanotopography of 20 × 10 mm mica sheets, untreated and treated with PDA, PDA + PEG, and PDA + PEG + EGCg. Measurements were performed using an NX10 Park AFM instrument (Park System, Suwon, Korea), equipped with 20-bit closed-loop XY and Z flexure scanners and a noncontact cantilever PPP-NCHR 5M. This instrument implements a True Noncontact ™ mode, allowing minimization of the tip–sample interaction, resulting in tip preservation, negligible sample nanotopography modification, and reduction of artifacts. On each sample, four different sample size areas were analysed (20 × 20, 10 × 10, 5 × 5, and 1 × 1 μm) at a scan rate of 0.1 Hz.

### 4.4. ζ-Potential Measurement

*ζ*-potential measurements were performed using SurPass 3, equipped with an adjustable gap cell (Anton-Paar GmbH, Graz, Austria). Measurements were conducted on 20 × 10-mm Ti plates (Ti foil 0.25 thickness from Sigma Aldrich), either uncoated or PDA, PDA + PEG, or PDA + PEG + EGCg coated. The streaming channel was created by adjusting the distance between the plate surfaces in the adjustable gap cell, using 100% of the total surface area for measurement. Measurements were performed in a 0.001-M KCl solution, according to pH scan method. This consists of the measurement of streaming current at different pH, between 2.5 and 9.5. The pH of the electrolyte solution was modified automatically by the instruments using a 0.05-mol/L HCl and 0.05-mol/L NaOH. At each pH point, three measurements were performed in order to condition the sample, then the fourth value was taken and reported. Streaming potential and streaming current measurements are known to be useful methods for the investigation of charge displacement in the electrical double layer caused by an external force shifting the liquid phase tangentially against the solid [41,42]. The *ζ*-potential value is referred to as the approach developed by Helmholtz and Smoluchowski:(1)$$\zeta =\frac{dIstr}{d\Delta p}\times \frac{n}{\varepsilon \times \varepsilon_{0}}\times \frac{L}{A}$$

The coefficient is the streaming current, *L* is the length of the slit channel formed between the two samples forming the capillary, and *A* is the area cross section of the capillary. *η* and *ε* are the viscosity and dielectric coefficient of the solution forced to pass through the capillary. If we use the Ohm’s law *Istr = Ustr*/*R* (where *R* is the electrical resistance inside the streaming channel and *Ustr* is the potential) Equation (2) becomes:(2)$$\zeta =\frac{dIstr}{d\Delta p}\times \frac{n}{\varepsilon \times \varepsilon_{0}}\times \frac{L}{A}\times \frac{1}{R}$$

However, if the solid sample contributes to the conductance inside the streaming channel (i.e., metals), the calculation of streaming potential provides an apparent value of zeta potential, because the measurement of the cell resistance, *R*, is affected. Furthermore, the application of the Helmholtz and Smoluchowski approach is possible with just the planar solids surface, as in our case, because an needs an exact knowledge regarding the geometry of the streaming channel, that is, the cell constant *L*/*A*. For the rectangular titanium slit, the length, *L*, and width are determined by the solid sample size; instead, the gap height is calculated from the measured volume flow rate of liquid passing through the streaming channel and the generated differential pressure. All calculations were performed by the instrument software.

### 4.5. Wettability

Wetting measurements were performed by a DCA 400 tensiometer (Gibertini, Milano, Italy), using a 60-μm/s stage speed, according to the Wilhelmy plate method. Briefly, a microbalance measures the force exerted on a solid (a square titanium plate, 2.4 cm a side, treated on both sides, in the present case) as it is immersed in a liquid (Milliq water in the present experiments). The total force can be written as:(3)$$F=W+\gamma_{L}\mathrm{pco}\vartheta +F_{b}\;$$
where *W* is the weight of the sample (i.e., its mass times the gravitational acceleration), *γ*_L_ is the surface tension of the wetting liquid, *p* is the sample perimeter, and *F*_b_ is the buoyancy. Because *W* is measured by the microbalance, *F_b_* can be zeroed by extrapolation to zero depth of immersion (ZOI), and *p* can be measured independently; it is possible to calculate the contact angle, *ϑ*, of a liquid of known surface tension, *γ*_L_, on a solid of known perimeter, *p*. The surface tension of water, in the present case, was measured by performing the same experiment on a freshly plasma-cleaned platinum sample (*ϑ* = 0, cos *ϑ* = 1), yielding 71.6 mJ/m^2^, in perfect agreement with expected result.

The experiments were performed by acquiring two wetting loops to check for hydration effects. Samples were first advanced 5 mm from ZOI, then withdrawn to zero depth (first wetting loop), then advanced 7 mm from ZOI and completely retracted (second wetting loop). Through the instrument software, the advancing and receding contact angles were calculated from the force measured at ZOI in the first wetting loop advancing and withdrawing, respectively.

### 4.6. Antioxidant Power of EGCg Coated Samples

The antioxidant power of coated Ti cylinders was evaluated through the widely-used DPPH (2,2-Diphenyl-1-picrylhydrazyl) method described by Brand-Williams et al. [43]. Using a colorimetric approach, this test measures the ability of the test solution to scavenge the DPPH radical.

Briefly, an aliquot of 2000 µL of water:ethanol 50:50 (*v*/*v*) solution was added to a sample uncoated as a blank, coated with PDA + EGCg and PDA + PEG + EGCg soaked for 30 min, 14 replicates for each condition. Separately, a DPPH solution (0.05 mg/mL *w*/*v*) in ethanol was prepared, and 2000 µL of this solution was added to the reaction mixture. Then the solution was shaken and incubated for 30 min at room temperature in the dark; the absorbance was recorded at 525 nm. Blank solution was constituted by a solution of water:ethanol alone. The percentage inhibition of the DPPH radical by the samples was calculated using the following equation:(4)$$\%Reduction=\left(\frac{A_{0}\times A_{1}}{A_{0}}\right)\times 100$$
where *A*_0_ is the absorbance of control sample and *A*_1_ is the absorbance of the test sample.

## 5. Conclusions

In conclusion, the presented results confirm the building up of a nanometre-thick, composite surface layer on top of PDA-coated titanium samples, endowed by anti-oxidant properties promoted by the EGCg molecular structure and by the hydrophilicity of the PEG repeating unit. The terminally linked aPEG chains enable a different interfacial environment around the antioxidant and potentially antimicrobial surface-linked EGCg as compared to plain EGCg coupling. Future experiments will be directed to understand whether the different interfacial nanoarchitecture affects relevant biological interactions, namely proliferation and tissue repair by soft tissue cells, bacterial attachment, and pathological biofilm formation.

## Figures and Tables

**Figure 1 ijms-24-02661-f001:**
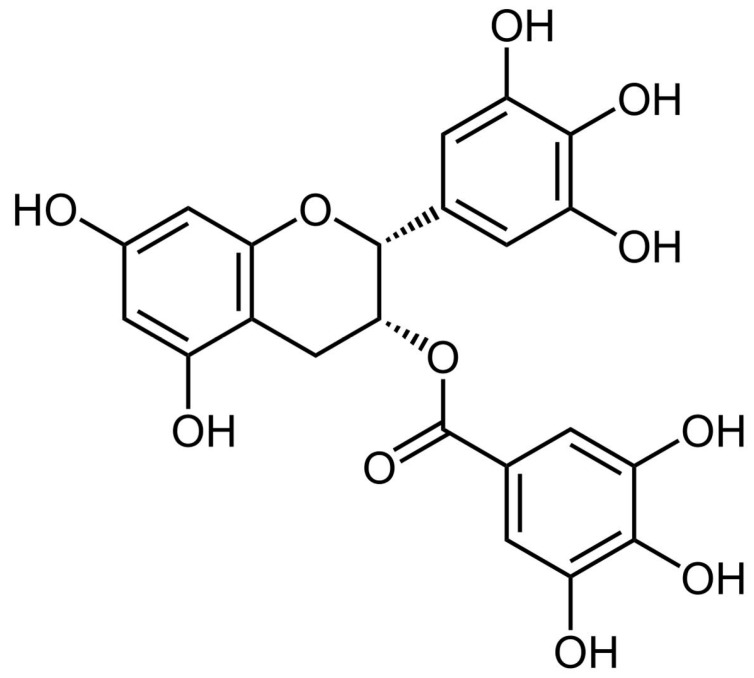
Epigallocathechin gallate (EGCg) molecular structure.

**Figure 2 ijms-24-02661-f002:**
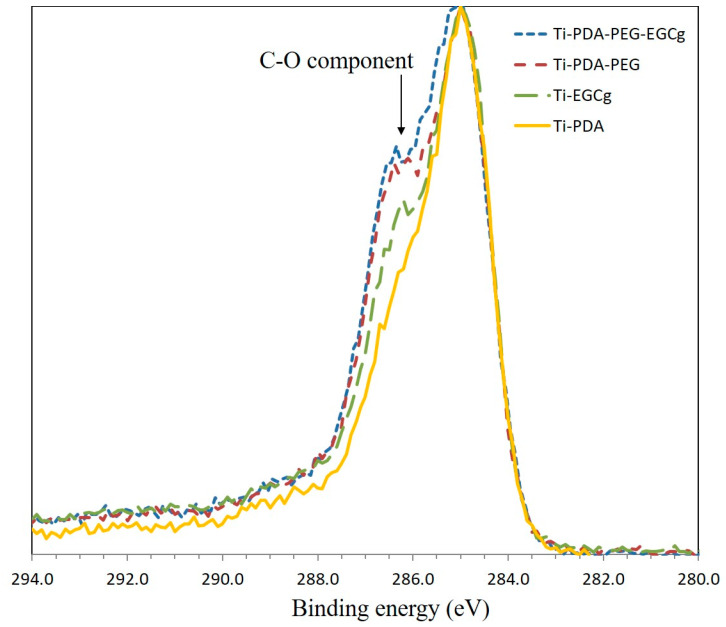
Normalized C1s peak of tested samples. The binding energy of all peaks has been referenced to the internal standard C-C component (285.0 eV). The figure shows the increasing contribution to the peak of the 286.5 eV component due to C-O containing moieties.

**Figure 3 ijms-24-02661-f003:**
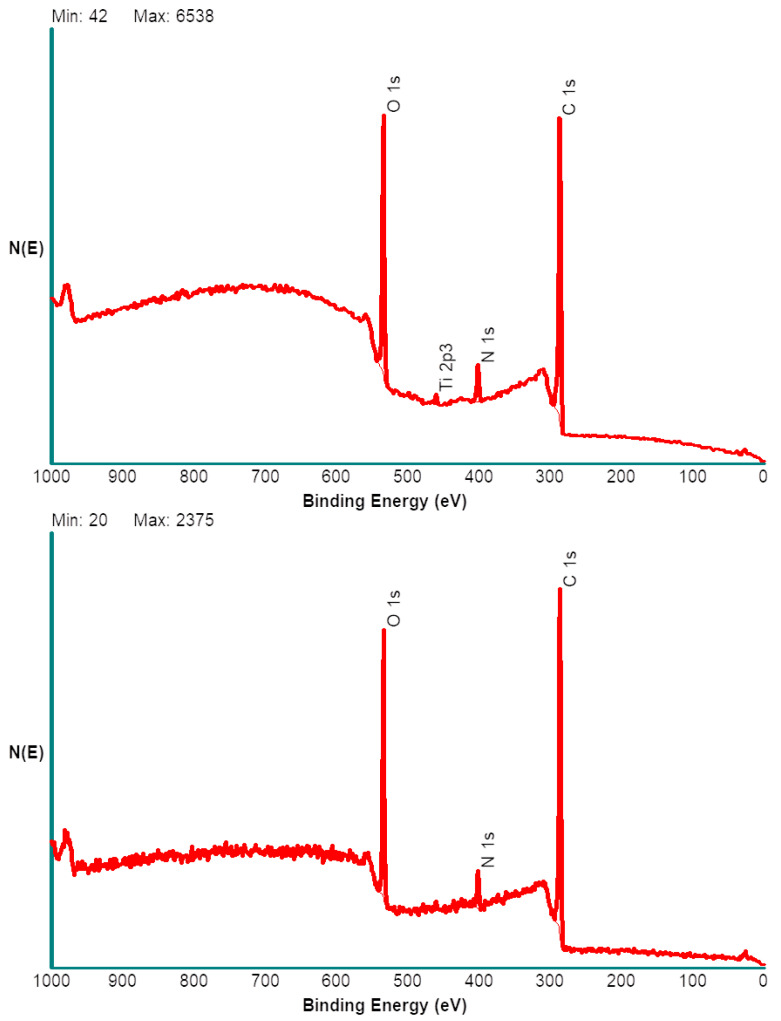
XPS spectra of Ti-PDA-aPEG-EGCg samples acquired at grazing (15°, bottom) and high (75°, top) take-off angle.

**Figure 4 ijms-24-02661-f004:**
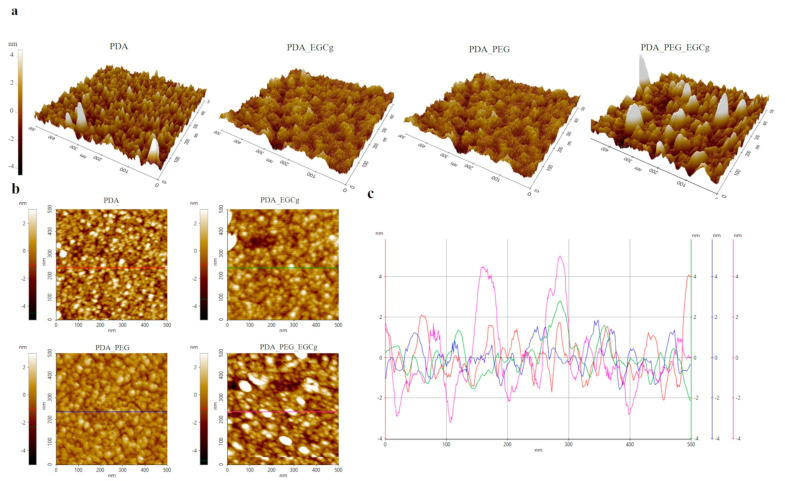
AFM analysis of mica coated with PDA, PDA-EGCg, PDA-aPEG, and PDA-aPEG-EGCg. (**a**) 3D images of 5 × 5 µm areas, (**b**) top view of 5 × 5 µm areas, and (**c**) line scan showing height profile.

**Figure 5 ijms-24-02661-f005:**
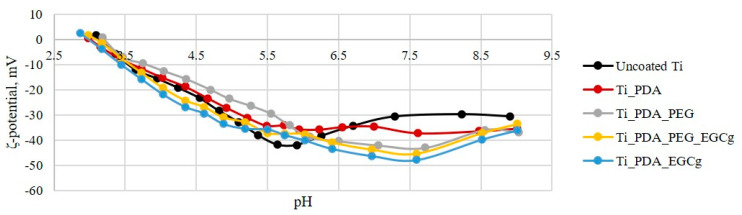
pH scan of Ti, Ti-PDA, Ti-PDA-aPEG, Ti-PDA-EGCg and Ti-PDA-aPEG-EGCg samples.

**Figure 6 ijms-24-02661-f006:**
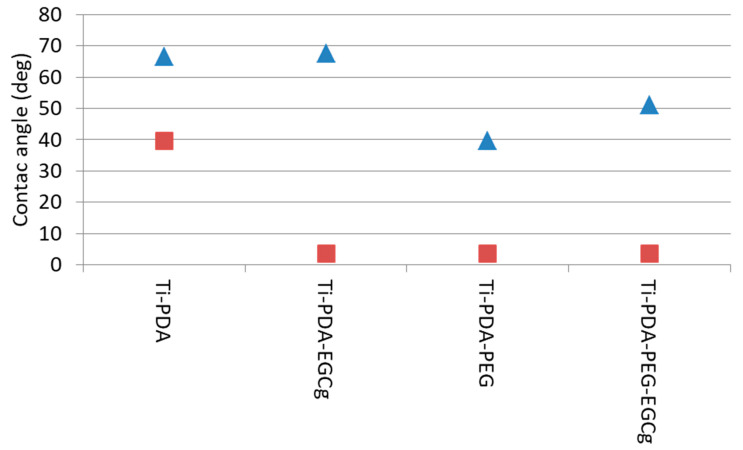
Measured advancing (triangle) and receding (square) angles on tested samples.

**Figure 7 ijms-24-02661-f007:**
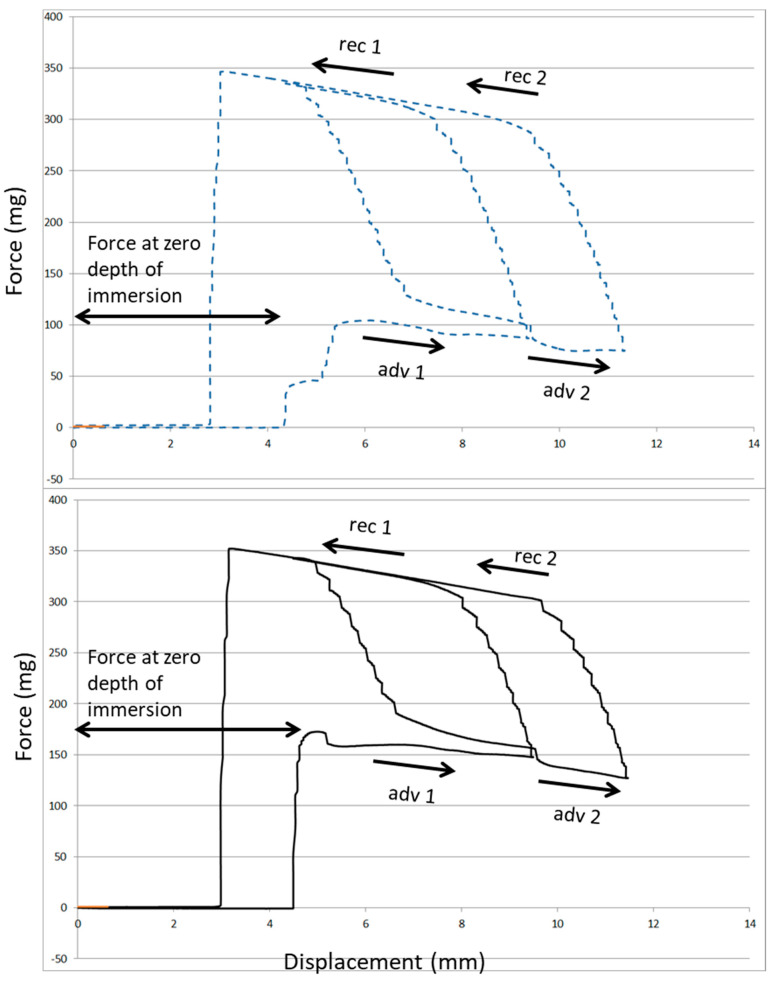
Wetting loops obtained by Wilhelmy plate measurements on Ti-PDA-EGCg (top) and Ti-PDA-aPEG-EGCg (bottom). Note the significantly higher (~50%) increase of the capillary force exerted by water on Ti-PDA-aPEG-EGCg, as indicated by the force at zero depth of immersion (ZOI). adv 1 = advancing portion of the first wetting loop; adv 2 = advancing portion of the second wetting loop; rec 1 = receding portion of the first wetting loop; rec 2 = receding portion of the second wetting loop.

**Table 1 ijms-24-02661-t001:** Surface composition (% at) of tested samples as detected by XPS analysis. Data show average and standard deviation, three replicates for each sample.

Sample	O	Ti	N	C	N/C	O/C
Ti	47.3 ± 0.3	19.6 ± 0.2	0.4 ± 0.1	32.6 ± 0.3	0.012	1.450
Ti-PDA	19.3 ± 0.6	0.3 ± 0.2	8.0 ± 0.6	72.4 ± 0.6	0.111	0.267
Ti-PDA-aPEG	20.4 ± 0.3	0.2 ± 0.2	6.5 ± 0.9	72.9 ± 0.8	0.090	0.279
Ti-PDA-EGCg	19.7 ± 1.1	0.3 ± 0.0	6.4 ± 0.3	73.7 ± 1.2	0.087	0.267
Ti-PDA-aPEG-EGCg	22.2 ± 1.1	0.7 ± 0.5	5.2 ± 0.6	71.8 ± 1.7	0.072	0.309

**Table 2 ijms-24-02661-t002:** Antioxidant power of Ti-PDA-EGCg, Ti-PDA-PEG-EGCg titanium cylinder, as measured by the DPPH test.

Sample	Reduction (%)
Ti-PDA-EGCg	13.3 ± 2.2
Ti-PDA-aPEG-EGCg	14.0 ± 2.8

**Table 3 ijms-24-02661-t003:** Surface roughness of mica and of mica after sequential coating steps, as expressed by vertical Sa (arithmetical mean height) and hybrid Sdr (developed interfacial area ratio) roughness parameters.

Sample	Sa (µm)	Sdr (%)
Uncoated	0.0001	0.1972
PDA	0.0009	4.1445
PDA-EGCg	0.0008	2.2751
PDA-aPEG	0.0007	3.2941
PDA-aPEG-EGCg	0.0012	8.8571

## Data Availability

The raw data supporting the data presented in this study are readily available on request from the corresponding author.

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
