# Peer review of "Engineering Interfacial Environment of Epigallocatechin Gallate Coated Titanium for Next-Generation Bioactive Dental Implant Components"

_ijms, 2023, doi:10.3390/ijms24032661_

Round 1

Reviewer 1 Report

This manuscript constructed a EGCg-based surface coating on the titanium samples to impart antioxidant and antimicrobial properties. The research can give guidance for related researchers. But there are still some more issues need to be considered. The details are listed as follows:

1. The quality of Figure 4 should be improved, such as the size of the values.

2. The information of Table 1 are too tanglesome. The results could be expressed as "avg ± std"

3. In Table 2, in addition to EGCg, PDA was also reported to possess antioxidant ability. The DPPH test result of other samples should be provided to study the effect of PDA.

4. There are some spelling errors, such as EGCc. Please check.

5. Why did the authors select EGCg to prepare coating? What are the advantages of EGCg compared to other polyphenols?

6. The author declared that the coating was expected to impart antioxidant and antimicrobial properties. Then why the antimicrobial property was not investigated?

7. There are some recent literature works about polyphenols or PDA that are relevant to this work (e.g., https://doi.org/10.1039/D2MH00768A and https://doi.org/10.1039/D2MH01151D), which can be considered to be cited.

Reviewer 2 Report

The authors reviewed research articles dedicated to the Engineering interfacial environment of epigallocatechin gallate coated titanium.

There are several problems to be addressed:

1. The authors should add the conclusion section 2. There are several publications related to this subject. Include few references from “International Journal of Molecular Sciences”.

Reviewer 3 Report

Dear author,

We have read and revised your manuscript describing the experiments of EGCg coating onto Titanium.

The concept sounds interesting and is well introduced. However, there are many points that should be reorganized, in order to clarify the manuscript.

> Title: please re-write the title, so that it expresses more the interest of EGCg coatings and the objective

> Abstract : please clarify and emphasize the main results.

> Materials and Methods : this section should be put directly after Introduction

> Results : please add Figures and Tables directly after they are cited in the Results (and not as a specific paragraph 'Tables and Figures'

> Atomic Force Microscopy: Why did you use a different substrate (Mica) for this experiment? It looses almost all its interest, as the objective is to create efficient coatings onto Titanium. Please provide AFM results of Ti coated (as for the other experiments : Bare Ti / Ti-PDA / Ti-PDA-EGC / Ti-PDA-PEG /Ti-PDA-PEG-EGC

> Wettability: in Fig.6, results of Bare Ti should be added (and a photograph of representative results might hep readability

Yours faithfully,

Round 2

Reviewer 1 Report

It can be accepted now.

Reviewer 3 Report

Dear Authors,

Thank you for sending a revised version of your manuscript.

The added modifications render the whole manuscript much easier to read and follow.

The only element that still have to be corrected is the writing in Table 1 (i.e 22,2 should be written 22.2)

The organization with Materials and Methods after Discussion looks surprising for IJMS, but if it is the Editor recommandation....

Best regards,